# The Gut-Ex-Vivo System (GEVS) Is a Dynamic and Versatile Tool for the Study of DNBS-Induced IBD in BALB/C and C57BL/6 Mice, Highlighting the Protective Role of Probiotics

**DOI:** 10.3390/biology11111574

**Published:** 2022-10-27

**Authors:** Romina Monzani, Mara Gagliardi, Nausicaa Clemente, Valentina Saverio, Elżbieta Pańczyszyn, Claudio Santoro, Nissan Yissachar, Annalisa Visciglia, Marco Pane, Angela Amoruso, Marco Corazzari

**Affiliations:** 1Department of Health Science & Center for Translational Research on Autoimmune and Allergic Disease (CAAD), University of Piemonte Orientale, 28100 Novara, Italy; 2Department of Health Science & Interdisciplinary Research Center of Autoimmune Diseases (IRCAD), University of Piemonte Orientale, 28100 Novara, Italy; 3Department of Health Science, Center for Translational Research on Autoimmune and Allergic Disease (CAAD), Interdisciplinary Research Center of Autoimmune Diseases (IRCAD), University of Piemonte Orientale, 28100 Novara, Italy; 4The Mina and Everard Goodman Faculty of Life Sciences, Bar-Ilan Institute of Nanotechnology and Advanced Materials, Bar-Ilan University, Ramat-Gan 5290002, Israel; 5Probiotical Research Spa, 28100 Novara, Italy

**Keywords:** IBD, UPR, ex vivo organ, DNBS, Gut-Ex-Vivo System, GEVS, ferroptosis, apoptosis, probiotics

## Abstract

**Simple Summary:**

IBD is considered a modern and western diet-related disease characterized by uncontrolled immune activation, resulting in chronic bowel inflammation and tissue damage. Although the precise causes of the onset of the disease are still elusive, it seems that both the environment, genetic predisposition and the dysregulation of the intestinal microbiota are actively involved. The development of a model to study the etiopathology of this disease characterized by an increasing incidence in the population is urgently needed. We have recently developed an organ-on-chip system (Gut-Ex-Vivo System, GEVS) to model IBD induced by DNBS in the colon of mice of the BALB/c strain. Here, we provide data demonstrating that the process can also be efficiently induced in mice of another strain, C57BL/6, which is usually less sensitive to this treatment, using our GEVS. Furthermore, we have shown that the system also replicates other characteristics of human pathology, such as the induction of the two most represented cell death pathways responsible for the tissue damage characteristic of IBD. Finally, we demonstrate that our system can be used efficiently to test new therapeutic interventions, such as those based on the use of probiotics. Indeed, we demonstrated the positive impact of both Lactobacilli and Bifidobacteria.

**Abstract:**

Background: IBD is a spectrum of pathologies characterized by dysregulated immune activation leading to uncontrolled response against the intestine, thus resulting in chronic gut inflammation and tissue damage. Due to its complexity, the molecular mechanisms responsible for disease onset and progression are still elusive, thus requiring intense research effort. In this context, the development of models replicating the etiopathology of IBD and allowing the testing of new potential therapies is critical. Methods: Colon from C57BL/6 or BALB/c mice was cultivated in a Gut-Ex-Vivo System (GEVS), exposed for 5 h to DNBS 1.5 or 2.5 mg/mL, in presence or absence of two probiotic formulations (P1 = *Bifidobacterium breve* BR03 (DSM16604) and B632 (DSM24706); P2 = *Lacticaseibacillus rhamnosus* LR04 (DSM16605), *Lactiplantibacillus plantarum* LP14 (DSM33401) and *Lacticaseibacillus paracasei* LPC09), and the main hallmarks of IBD were evaluated. Results: Gene expression analysis revealed the following DNBS-induced effects: (i) compromised tight junction organization, responsible for tissue permeability dysregulation; (ii) induction of ER stress, and (iii) tissue inflammation in colon of C57BL/6 mice. Moreover, the concomitant DNBS-induced apoptosis and ferroptosis pathways were evident in colon from both BALB/c and C57BL/6 mice. Finally, the co-administration of probiotics completely prevented the detrimental effects of DNBS. Conclusions: Overall, we have provided results demonstrating that GEVS is a consistent, reliable, and cost-effective system for modeling DNBS-induced IBD, useful for studying the onset and progression of human disease at the molecular level, while also reducing animal suffering. Moreover, we have confirmed the beneficial effect of probiotics administration in promoting the remission of IBD.

## 1. Introduction

Gut inflammation is a feature shared by several human diseases specifically affecting the gastrointestinal tract, such as celiac disease [1], cystic fibrosis [2] and inflammatory bowel disease [3]. However, mounting evidence points to the role of inflammation in the pathogenesis of other human diseases, which apparently do not directly involve the gut, such as Parkinson’s disease [4] and type 2 diabetes [5]. Moreover, although the role of microbiota in gut inflammation associated with the above-mentioned diseases is still unclear, dysbiosis has been documented in most patients and might potentially represent a new frontier in the clinical treatment of those pathologies.

In this scenario, the availability of tools to model the pathogenesis of such diseases is a critical point. Although tissues/cells directly derived from patients represent the best option for molecular studies, to test new potential therapeutic agents and design a personalized therapy, their availability and limited survival in vitro often represent the main limitations. 

The most widely used models in these studies are 2D cell cultures, which are the most available, cost effective and easy to use models. However, they are characterized by several limitations due to the complete lack of tissue architecture and typical cell–cell and cell–ECM communication. Moreover, they usually involve the use of immortalized or transformed cells, which potentially impacts on results [6]. An evolution of these models is represented by the 3D cell cultures, such as spheroids and organoids. While the first model can provide some characteristics of physiological conditions such as the 3D structure and the deposition of ECM, it also represents a too simple model compared to a tissue/organ condition, although cost effective and relatively easy to generate and manipulate [7]. On the other hand, organoids represent a consistent step forward in the generation of a model to mimic its corresponding in vivo tissue/organ [8]. However, organoids are difficult to generate, time consuming, expensive, and difficult to standardize [6]. Animal models are major source for studying human disease pathologies, with mice representing the most used species. They indubitably represent the best model when an animal model can be generated to recreate human pathology, the latter thus representing the main limitation. Of note, other limitations are the obvious physiological and morphological differences between humans and mice. This is why not all diseases can be efficiently modeled in mice. In this context, several animal models and protocols have been developed to study and to mimic IBD, such as those using dextran sulfate sodium (DSS) [9], 2,4,6-trinitrobenzenesulfonic acid (TNBS) [10] or dinitrobenzene sulfonic acid (DNBS) [11]. These models well replicate the pathogenesis of human IBD and are widely used, although they are not without animal suffering. Similarly, to other animal models, they also have a number of disadvantages including the overall length of experiments, and costs of animal purchase and maintenance. Moreover, animal models are intrinsically subjected to biological variations based on both genetic and nongenetic components. Indeed, even genetic littermates are not identical and can be differently affected by environmental conditions and social hierarchy. Altogether, these variations affect experimental results, but their impact can be limited increasing the overall number of animals included in each research group. Therefore, in the context of a continuous and increased demand in the reduction in the number of animals in scientific experimentation and reduction of animal suffering, the development of alternative models is required.

We recently described an evolution of an organ on chip model originally developed by Yissachar and colleagues [12], a Gut-Ex-Vivo System (GEVS) to study both celiac disease [13] and IBD [3], providing evidence showing the consistency and reliability of the system, together with reduced numbers and suffering of animals. In particular, we demonstrated the ability of the system to model the pathogenesis of DNBS-induced IBD in colon from BALB/c mice.

Here, we show that GEVS can also be used to model DNBS-induced IBD pathogenesis in colon from C57BL/6 mice, and that both apoptotic and ferroptotic cell death pathways characteristic of the human pathology are replicated in both animal strains. Moreover, we also show that the system can be used efficiently to test new therapeutic approaches such as those based on the use of probiotics, unveiling the molecular mechanism(s).

## 2. Materials and Methods

### 2.1. Reagents and Materials

DNBS (2,4-Dinitrobenzenesulfonic acid hydrate; CAS Number: 698999-22-3), poly(dimethylsiloxane) (PDMS; Sylgard 184 Elastomer base) was obtainede from Merck (Burlington, VT, USA); IMDM (Iscove’s Modified Dulbecco’s Medium), KnockOut serum replacement, B-27, N-2 supplements, L-glutamine, non-essential amino acids (NEAA) and HEPES were obtained from Gibco (Grand Island, NY, USA); Dulbecco’s Modified Eagle’s Medium (DMEM), fetal bovine serum (FBS) and penicillin/streptomycin were obtained from Euroclone (Milano, Italy); TripleXtractor was obtained from GRiSP (Porto, Portugal); ExcelRT Reverse Transcriptase and Excel-Taq FAST qPCR SybrGreen were obtained from Smobio (Hsinchu, Taiwan); and oligonucleotide based primers were obtained from IDT (Coralville, IA, USA).

### 2.2. Silicone-Based Device and Colon Culture

Colons were freshly isolated from 13-day-old BALB/c mice and cultivated in a silicone-based GEVS (Gut Ex-Vivo System), as previously described [3,13]. Briefly, each device consists of six parallel isolated chambers to contain up to six samples of mouse colon. Colon lumens were infused with a complete medium, through a coordinated infusing–drying pump. Each chamber was filled with complete medium (outer medium) to maintain tissue viability, in a static condition. A lab hot plate was used to maintain a temperature of 37 °C, during each experiment, while a mixture of 5% CO_2_ and 95% O_2_ was provided to the device from a compressed gas cylinder [3,12,13].

### 2.3. Colon Cultures and Treatments

Each colon section was infused with serum-free tissue culture medium containing IMDM, supplemented with 20% KnockOut serum replacement (Gibco, Grand Island, NY, USA), 2% B-27 and 1% of N-2 supplements, 1% L-glutamine, 1% NEAA, 1% HEPES and stimulated with DNBS (1.5–2.5 mg/mL).

The tissue culture medium was loaded into a 5 mL syringe and infused into the device input ports by a syringe pump (flow rate of 99 ul/h) [3,13].

### 2.4. Probiotics Formulations and Treatments

P1 and P2 probiotic formulations were provided by PROBIOTICAL S.p.a. (Novara, Italy), with P1 containing *Bifidobacterium breve* BR03 (DSM16604) and B632 (DSM24706) at 2 × 10^9^ AFU/g, while P2 consisting of *Lacticaseibacillus rhamnosus* LR04 (DSM16605), *Lactiplantibacillus plantarum* LP14 (DSM33401) and *Lacticaseibacillus paracasei* LPC09 (DSM24243), at 3 × 10^9^ AFU/g [1]. Both formulations were resuspended in PBS and administered as described below.

P1 or P2 were diluted in complete medium at 0.2 × 10^9^/mL or 0.6 × 10^9^/mL of P1 or P2 bacterial cells, respectively, and administered alone or in combination with DNBS (1.5 mg/mL).

### 2.5. Quantitative PCR (qPCR)

Total RNA was isolated by using TripleXtractor reagent and ExcelRT Reverse Transcriptase was used to produce cDNA, by using 2 μg of total RNA. Quantitative PCR (qPCR) reactions were performed by using the Excel-Taq FAST qPCR SybrGreen and a CFX96 thermocycler (Bio-Rad, Hercules, CA, USA). Primer sequences were designed by using the online IDT PrimerQuest Tool software (IDT; https://eu.idtdna.com/Primerquest/Home/Index, accessed on 15 September 2021), and sequences are reported below (Table 1).

The GAPDH mRNA level was used as an internal control, and the comparative Ct method (ΔΔCt) was used for relative quantification of gene expression [14].

### 2.6. Western Blotting Analysis

Colon tissues were lysed by using the Cell Lytic Buffer supplemented with both proteases (PIC, Merck) and phosphatases (NaF 10 mM; Na_3_VO_4_ 1 mM) inhibitors. Equal amounts of protein lysates (20 mg) were subjected to SDS-Page separation and proteins electroblotted onto Nitrocellulose (Merck) membranes. Non-fat 5% dry milk (Merck) in PBS was used as a blocking solution, and indicated primary antibodies, in blocking solution were incubated o.n. at 4 °C. Appropriate HRP-conjugated secondary antibodies were diluted in blocking solution (1:5000) and incubated for 1 h at r.t. A Westar ANTARES ECL kit (Cyanagen, Bologna, Italy) was then used and the signal acquired by a ChemiDoc^TM^ Touch (Bio-Rad), and analyzed using Image Lab software (5.0; Bio-Rad). Primary antibodies were as follows: anti-Bip/Grp78 (1:500; Santa Cruz, Dallas, TX, USA); anti-Calnexin (1:500; Santa Cruz); anti-Calreticulin (1:1000; Abcam, Cambridge, UK); anti-PARP cleaved (1:500; Cell Signaling, Danvers, MA, USA); anti-SLC7A11 (1:250; Cell Signaling); anti-Occludin (1:500; Cell Signaling); anti-Caspase-3 (1:250; Santa Cruz); anti-aActin (1:5000; Sigma, Darmstadt, Germany); anti-bTubulin (1:5000; Santa Cruz). HRP-conjugated secondary antibodies were diluted in blocking solution (1:5000; Jackson ImmunoResearch, Cambridgeshire, UK).

### 2.7. Immunofluorescence Analysis

At the end of the treatments, colon tissues were removed from GEVS, washed twice with cold PBS and immediately embedded in OCT and frozen at −80 °C. Tissue slices of 8–10 mm, on appropriate glass slides, were washed with cold PBS and fixed with 4% PFA for 24–48 h at 4 °C. Sections were permeabilized with 0.5% Triton X-100 in cold PBS for 24 h at 4 °C. Donkey serum (10%) plus 0.05% Triton X-100 in cold PBS was used as blocking solution (2 h at 4 °C). Appropriate primary antibodies were diluted in 1% donkey serum plus 0.05% Triton X-100 in cold PBS, then incubated o.n. at 4 °C. Appropriate fluorescently conjugated secondary antibodies were diluted at 1:1000 in 1% donkey serum plus 0.05% Triton X-100 in cold PBS and incubated for 2 h at 4 °C. ProLong™ Gold Antifade with DAPI mounting solution (ThermoFisher, Waltham, MA, USA) was used to preserve sections, together with a glass coverslip. Nail polish was used to seal slides. Images were acquired by using an SP8 LIGHTNING Confocal Microscope (Leica, Wetzlar, Germany) and images analyzed by LAS X Life Science (Leica) software. Primary and secondary antibodies were as follows: anti-E-Cadherin (1:250; Santa Cruz); Alexa Fluor 488 donkey anti-Rat (1:500; Invitrogen, Waltham, MA, USA); anti-ZO-1AF647 (1:200; Santa Cruz, Dallas, TX, USA).

### 2.8. ELISA

IFNg, TNFa and IL-10 were measured in colon tissue lysates by using Mouse IFN-gamma Quantikine, Mouse TNF-alpha Quantikine and Mouse IL-10 Quantikine ELISA kits (Bio-Techne, Minneapolis, MN, USA), as recommended by the supplier. Optical densities were analyzed by a SPARK Multimode Microplate Reader (TECAN, Männedorf, Switzerland). Values were normalized to total protein concentration, as previously reported [1,3,13].

### 2.9. Statistical Analysis

Experiments were performed in triplicate and repeated at least three times, and statistical analysis was performed using GraphPad software (GraphPad Software; GraphPad Prism 6). Student’s *t*-test or ANOVA were used to determine statistical significance.

A *p*-value of equal to or less than 0.05 was considered significant. mRNA expression levels were represented as ‘fold change over control’, r.l. relative levels. Histograms represent mean ± SD; **** *p* < 0.0001; *** *p* < 0.001; ** *p* < 0.01; * *p* < 0.05; ns non-significant.

## 3. Results

We recently demonstrated the possibility of using an organ ex vivo system (Gut-Ex-Vivo System or GEVS) in parallel or as an alternative to animal models to study the onset and progression of human diseases characterized by gut inflammation, such as celiac disease or IBD [3,13]. Indeed, in the case of IBD we designed a method consisting of the isolation of colon from BALB/c mice, and cultivation in our GEVS in the presence of DNBS (Figure 1).

We demonstrated that 5 h of exposure to DNBS is sufficient to (i) deregulate colon permeability through tight junction (TJ) protein expression deregulation, (ii) induce an ER stress response, (iii) up-regulate pro-inflammatory cytokines and down-regulate anti-inflammatory cytokines, resulting in inflammation and tissue damage, all typical ulcerative colitis (UC) signs [3].

We therefore, decided to expand the use of our GEVS to study the DNBS-induced UC in colon derived from the C57BL/6 mouse strain, which is notoriously less sensitive compared to the widely used BALB/c strain. We also evaluated the ability of our system and method to replicate the two main cell death pathways known to be responsible for epithelial cell demise upon UC onset/progression, such as apoptosis [15] and the recently described ferroptosis [16].

### 3.1. DNBS Triggers UC in the Colon of C57BL/6 Mice by Using a Gut-Ex-Vivo System (GEVS)

Colon samples freshly explanted from C57BL/6 mice were cultivated in a Gut-Ex-Vivo System as previously described [3]. UC was induced by exposing organs to 0, 1.5 or 2.5 mg/mL of DNBS, in the infusing medium, for 5 h. Tissue barrier efficiency was evaluated by measuring the expression of TJ proteins. Data reported in Figure 2 clearly show an imbalanced expression of these factors, as evidenced by down-regulated occludin (Figure 2A and Figure 7D) and up-regulated expression of both claudin-2 (Figure 2B) and claudin-15 (Figure 2C), in a dose-dependent manner [1,3,13], thus indicating a compromised barrier function. These data perfectly overlap those previously obtained by using colon from the BALB/c mouse strain [3]. We also observed a down-regulation of E-Cadherin expression at both mRNA and protein levels (Appendix A).

Next, we evaluated the ability of DNBS to stimulate the ER stress response, as previously evidenced in colon from BALB/c mice, in the same experimental conditions [3]. To this aim, the expression of the well-characterized ER stress markers ATF4, ATF6 and XBP1s were evaluated at the mRNA level. Data shown in Figure 3A indicate a DNBS-dependent induction of ER stress, in a dose-dependent manner. These data were also confirmed by western blotting analysis of specific ER stress-related proteins such as BIP/Grp78, Calreticulin (CRT) and Calnexin (CNX) [17], whose expression increased in colon tissues exposed to DNBS (1.5 mg/mL; Figure 3B).

Finally, the expression of the pro-inflammatory cytokines IFNγ and TNFα together with the anti-inflammatory IL-10 were evaluated at both the mRNA and protein levels, in the same experimental conditions. As reported in Figure 4, a clear dose-dependent up-regulation of pro-inflammatory and down-regulation of anti-inflammatory cytokines was evident in tissues exposed to DNBS, compared to untreated controls.

Collectively, these data are similar to those previously obtained by stimulating BALB/c derived colon tissues with DNBS, in the same experimental conditions [3].

### 3.2. Apoptotic and Ferroptotic Cell Death Modalities Associated with DNBS-Induced UC Can Be Studied by Using a GEVS, in Colon from Both C57BL/6 and BALB/c Mice

Ulcerative colitis is characterized by progressive colonic epithelial cell demise due to consistent cell death induction, mainly through the stimulation of the apoptotic process [18]. To verify the induction of apoptosis in colon cultivated in our Gut-Ex-Vivo System and exposed to DNBS (5 h), we evaluated the expression of key pro-apoptotic markers such as BAX, PUMA and NOXA [19,20,21], at the mRNA level. Data reported in Figure 5 show a clear induction of the apoptotic markers described above, in a dose-dependent manner, and perfectly overlapping in the two mouse strains. Moreover, the activation of caspases, revealed by the progressive cleavage of the substrate PARP, confirmed the DNBS-stimulated apoptosis induction, in a dose-dependent manner (Figure 4, right panels).

Furthermore, it has been proposed that a recently described new form of cell death called ferroptosis may contribute to the demise of epithelial cells associated with UC [16]. To verify the involvement of this cell death signaling pathway in colon exposed to DNBS in our GEVS, we evaluated the expression of key ferroptotic markers. Indeed, we observed a dose-dependent up-regulation of the early ferroptotic markers such as CHAC1 [22,23] and PTGS2 [24], together with the downstream markers ACSL4 [25] and TRF1 [26], in tissues from both C57BL/6 (Figure 6, upper panels) and BALB/c (Figure 6, bottom panels) mice, with no major differences.

### 3.3. Probiotic Administration Prevents IBD Onset

Currently, there are no standard medical interventions to cure Crohn’s disease (CD) or ulcerative colitis (UC), the two main forms of IBD. Moreover, those clinical treatments able to reduce/inhibit the main features of IBD are often associated with the onset of side effects, thus limiting the treatment [27] and pushing researchers to look for alternative treatments or supplements to improve remission in IBD. Recently, probiotic-based therapy is emerging as an interesting adjuvant or alternative to canonical intervention, mostly due its relatively safety [28].

From this perspective, we evaluated the impact of probiotics on DNBS-induced UC in colon tissues from C57BL/6 mice cultivated in our GEVS. To this aim, we used two probiotic formulations composed of 2 Bifidobacteria (P1) or 3 Lactobacilli (P2), as described in the methods section. Therefore, colon was exposed to complete medium (CTRL), DNBS, P1 or P2 alone or a combination of DNBS+P1 or DNBS + P2 for 5 h. At the end of treatments, tissues were lysed for western blotting/ELISA analysis or were embedded in OCT for immunofluorescence analysis.

Data reported in Figure 7 clearly show that both P1 and P2 completely abolished the DNBS-stimulated production of the pro-inflammatory cytokines IFNg (Figure 7A) and TFNa (Figure 7B), and the down-regulation of the anti-inflammatory cytokine IL-10 (Figure 7C).

Moreover, we also observed the abolition of both ferroptotic and apoptotic cell death, in the same experimental conditions, as evidenced by the restored physiological levels of the ferroptotic marker SLC7A11 and the prevented activation of caspase-3 (Figure 7C, IB: SLC7A11 and IB: Pro-Casase-3, respectively).

Finally, our analysis also evidenced the ability of the two probiotic formulations to prevent the DNBS-induced deregulation of TJ proteins, as evidenced by the restored expression of occludin (Figure 7C, IB: OCL) and both expression and cellular localization of ZO-1 (Figure 8 and Appendix A).

Notably, our analysis also provided some evidence indicating the potential molecular mechanism by which these probiotic formulations might counteract the pathogenic effect of DNBS. Indeed, data reported in Figure 7D indicate that both the Bifidobacteria and Lactobacilli mix sustain the permeability of tissues, increasing the expression of TJ proteins, as evidenced by the increased expression of occludin in the presence of P1 or P2 alone (Figure 7D, IB: OCL, compare lanes 3 and 4 with lane 1, and OD analysis), while no effect was observed in the basal production of either anti- and pro- inflammatory cytokines (Figure 7C, Figure 7A and Figure 7B, respectively), in the same experimental conditions, thus indicating no immunomodulatory effects. Moreover, our analysis also revealed that probiotic formulations consistently inhibited both the apoptotic (as evidenced by the abrogated activation of caspase-3; Figure 7C IB: Pro-Caspase-3, compare lane 2 with lanes 5 and 6) and ferroptotic (as evidenced by the abrogated decrease of SLC7A11 protein level; Figure 7C IB: SLC7A11, compare lane 2 with lanes 5 and 6) by DNBS, thus sustaining the integrity and viability of tissues.

## 4. Discussion

IBD is characterized by gut inflammation, tissue dysregulation and damage, representing a complex multifactorial systemic disease with both genetic and environmental components being involved in the onset and progression [29]. Due to its complexity, its etiology at the molecular level and mechanisms are still elusive and under intensive investigation. In this context, although biopsies directly derived from affected individuals remain the gold standard for research, also potentially ensuring a personalized clinical intervention, the very small amount of sample together with its limited viability require the development of alternative approaches. Although cell lines are widely used, they are intrinsically limited since they completely lack the complexity and physiology of tissues. Of note, several animal models have been described and are currently used in this field of research, such as mice exposed to DSS or DNBS, which well replicate the onset of the disease. However, animals are expensive and experiments require time. Moreover, the above-mentioned models are not without suffering for animals.

We recently described an organ ex vivo culture of colon from 13–15-day-old mice, which can reproduce both the morphological and molecular features of IBD induced in mice exposed to DNBS. In particular, we reported that colon from BALB/c mice can efficiently be stimulated for 5 h with a range of concentrations of DNBS, ranging from 0.5 to 2.5 mg/mL, to induce the main hallmarks of IBD, such as decreased tissue permeability and altered morphology, induction of pro-inflammatory cytokines, and down-regulation of IL-10, in a dose-dependent manner. Interestingly, we also found the induction of ER stress, which has recently been associated with IBD onset and progression in patients, although its precise role is still elusive [30].

Here we reported the induction of IBD in colon from 13-day-old mice from a different strain, the C57BL/6. This is important because this strain has been described to be somewhat resistant to DNBS stimulation in vivo, thus requiring a higher dose of drug [31,32]. Indeed, we found overlapping induction of the main IBD markers in both C57BL/6 and BALB/c mice, by using DNBS at both 1.5 and 2.5 mg/mL. Moreover, we confirmed the induction of ER stress, a pro-inflammatory and stress-mediated signaling pathway, recently described as potentially involved in the etiology of IBD [33,34]. Although its precise role is still elusive, and its activities apparently contrasting, both sustaining cell survival and inducing cell demise through apoptosis induction [19,35], it might represent a new potential therapeutic target. Further studies are therefore required to reveal the role of ER stress in the pathogenesis of IBD. In this context, the organ ex vivo system (GEVS) we provided represents a useful tool to explore this hypothesis at the molecular level, together with the opportunity to test molecules/probiotics to inhibit inflammation and to restore physiological tissue conditions.

Cell demise responsible for gut tissue dysfunction and damage associated with prolonged inflammation of gut tissues upon IBD onset and progression has been reported to rely mainly on apoptosis execution [18]. We previously confirmed the induction of apoptosis in BALB/c colon exposed to DNBS by using our GEVS at both the morphological and molecular levels [3]. Here we provide data confirming these results in BALB/c mice-derived colon at the molecular level, as evidenced by dose-dependent up-regulation of pro-apoptotic factors such as BAX, PUMA and NOXA, together with the activation of caspases, the main apoptosis executioners. Importantly, we have provided data showing that the same pro-apoptotic pathway is also induced in C57BL/c in the same experimental conditions, reinforcing the concept related to the involvement of apoptosis in IBD, and also providing evidence of versatility and reliability of our organ ex vivo culture model.

The involvement of a new form of cell death, ferroptosis, which participates in the pathogenesis of IBD in patients, has also recently been described [16]. This is a non-apoptotic cell death modality that relies on the generation and intracellular accumulation of lipid peroxides, resulting in cell death [23]. Our results, obtained by using the Gut-Ex-Vivo System—GEVS—indicate the involvement of this new form of cell death in colon from both BALB/c and C57BL/6 mice exposed to DNBS, in a dose-dependent manner. We believe this has two main consequences: on one hand, it confirms the involvement of this form of cell death in the pathogenesis of IBD, and on the other hand it further confirms that our system is able to mimic the onset and progression of the pathology observed in patients, offering a further tool for research.

As previously mentioned, the incidence of IBD is continuously increasing in western communities, while an effective cure is not yet available to treat these patients. There seem to be several reasons for this delay in identifying a cure, including the high heterogeneity of the causes and triggers, some of which are still unknown. Moreover, some of the therapies currently in use, such as those based on the inactivation of TNFα, are not effective in all patients, and/or are associated with the onset of important systemic side effects, that limit their use [36]. Therefore, alternative therapy or supplements to improve remission of IBD are urgently needed. In this regard, the interest in microbiota-based therapeutic approaches is emerging for two main reasons: (1) their relative safety and few adverse effects compared to conventional treatments, and (2) the appearance of dysbiosis in IBD affected patients. In particular, a consistent reduced biodiversity paralleled by a reduction of *Lactobacilli* and *Bifidobacteria* spp. and increased *Bacteroides*, *Enterococci* and *Escherichia* spp. have been observed in patients affected by IBD compared to healthy people [37]. Therefore, rebalancing the microbiota composition, by means of probiotics administration, such as those consisting of Lactobacilli and/or Bifidobacteria, could offer new therapeutic opportunities. Although the mechanism(s) of action of probiotics seems to be strain specific, their anti-inflammatory activity is partly based on the following: (i) the production and release of short chain fatty acids (SCFA) with an immunomodulatory activity [38]; (ii) enhancing the mucus layer [39]; (iii) sustaining the expression of TJ proteins such as ZO-1, OCL and Claudins [40]; and (iv) stimulating the production of anti-inflammatory cytokines such as IL-10 and TGFb [41]. In this respect, we observed that two probiotic formulations based on three Lactobacilli (*Lacticaseibacillus rhamnosus* LR04 (DSM16605), *Lactiplantibacillus plantarum* LP14 (DSM33401) and *Lacticaseibacillus paracasei* (LPC09)) and two Bifidobacteria (*Bifidobacterium breve* BR03 (DSM16604) and B632 (DSM24706)) mix, respectively, completely reversed the detrimental effects of DNBS exposure of colon tissues, thus pointing out the importance of their immunomodulatory and anti-inflammatory activity with respect to this complex pathology. In particular, we observed that both formulations prevented the DNBS-stimulated inflammation of colon tissues, while sustaining both tissue integrity and permeability. Moreover, our data also confirmed the positive effects of probiotics on the expression of TJ proteins, as previously observed in patients [40], thus further confirming the ability of our system to mimic the pathology and to represent an excellent model to test new therapeutic approaches. Further studies are however required to uncover the precise molecular mechanism(s) of action of these microbiota-based probiotic formulations.

## 5. Conclusions

The pathogenesis of IBD is still elusive, with signs and progression that may vary depending on individuals. Further molecular studies are thus urgently required to define the precise sequence of events leading to the onset of the disease. In this regard, the availability of models that replicate well the main signs observed in patients is essential. Indeed, although the production of key pro-inflammatory cytokines and the downregulation of anti-inflammatory ones, together with dysregulated/relocated TJ proteins, represent typical hallmarks of IBD in affected patients, the chronology of their appearance, how/whether they are linked and their real impact is still unclear. Chemically-induced IBD animal models, together with our ex vivo model (GEVS) replicate well the appearance of those signs, although the concept of ‘who controls who’ is sometimes misleading [42,43,44,45]. Here we demonstrated that our GEVS can efficiently be used to dissect these mechanisms at the molecular level.

In the last few years, dysregulated ER function, thus leading to ER stress and triggering the unfolded protein response (UPR), is emerging as a new sign observed in patients affected by IBD, although its role is unclear [33,46,47]. Results from our ex vivo model demonstrate that we can mimic this event with our platform, which can help in understanding the real impact of ER stress in the pathogenesis of IBD, by using modulators of the UPR.

It is well documented that apoptotic cell death has been observed in epithelial cells at the inflamed sites in patients affected by IBD. Furthermore, induction of epithelial cell apoptosis has also been evident in several animal models mimicking IBD [48]. Moreover, ferroptosis has been implicated in both clinical UC patients and in murine experimental models, with significant modulation of typical ferroptotic markers. Indeed, administration of ferroptosis inhibitors significantly reduced IBD markers in mice exposed to DSS, pointing to the role of ferroptosis in the etiology of IBD [49]. Again, the possibility of using a flexible, coherent and reproducible model, such as the GEVS, will allow even deeper investigation of the molecular mechanisms underlying the induction of these cell death processes, in order to identify and test new potential therapeutic targets.

In conclusion, our results clearly indicate a strong consistency of results obtained by our organ ex vivo culture system (GEVS), in terms of variability of results from different animals. This aspect is important when working with animals, as it has a significant impact on the determination of numbers required within each group of animals during the experimental design. Our approach therefore allows a significant reduction in the total number of animals included in a study. It also allows the following: (i) reduction of experimental costs, (ii) reduction of the duration of experiments, (iii) good potential in the study of events at the molecular level in order to (iv) better clarify the pathogenesis of IBD, (v) identify/test new potential therapeutic targets (such as probiotics) and (vi) reduce animal suffering. This last point is becoming of great interest, in view of the increasing demand, by the scientific community, for a constant reduction in numbers and animal suffering.

## Figures and Tables

**Figure 1 biology-11-01574-f001:**
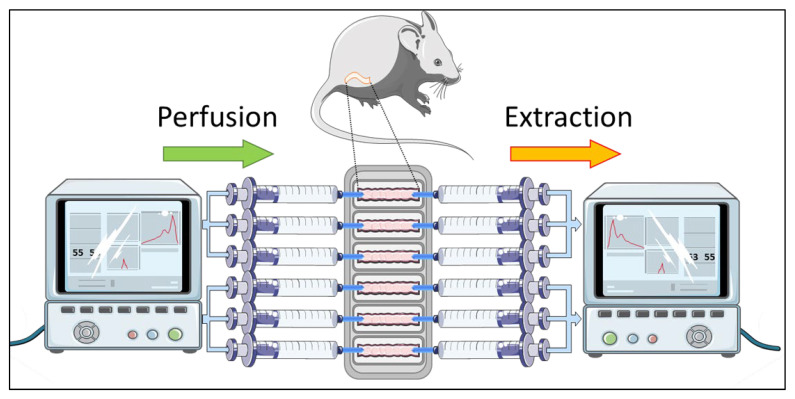
Gut-Ex-Vivo System—GEVS—and IBD induction. Schematic representation of a GEVS, in which mouse colon is cultivated in a dynamic condition and stimulated for 5 h with DNBS, to stimulate the onset of UC [3].

**Figure 2 biology-11-01574-f002:**
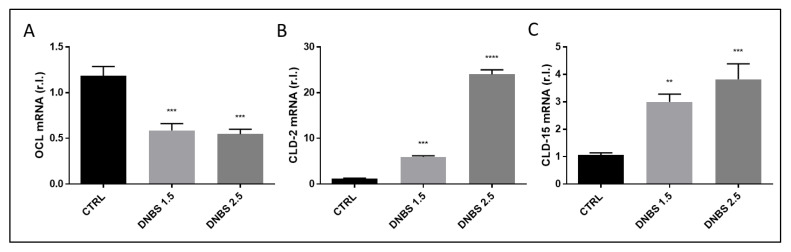
DNBS-induced impaired tissue permeability. Colon from C57BL/6 was cultivated for 5 h in a GEVS and untreated (CTRL) or treated with the indicated concentrations of DNBS and tissue permeability was evaluated by measuring the mRNA levels of tight junction (TJ) components, (**A**) Occludin (OCL), (**B**) Claudin-2 (CLD2) or (**C**) Claudin-15 (CLD15), by qPCR. Histograms represent mean ± SD of triplicate sample; **** *p* < 0.0001; *** *p* < 0.001; ** *p* < 0.01.

**Figure 3 biology-11-01574-f003:**
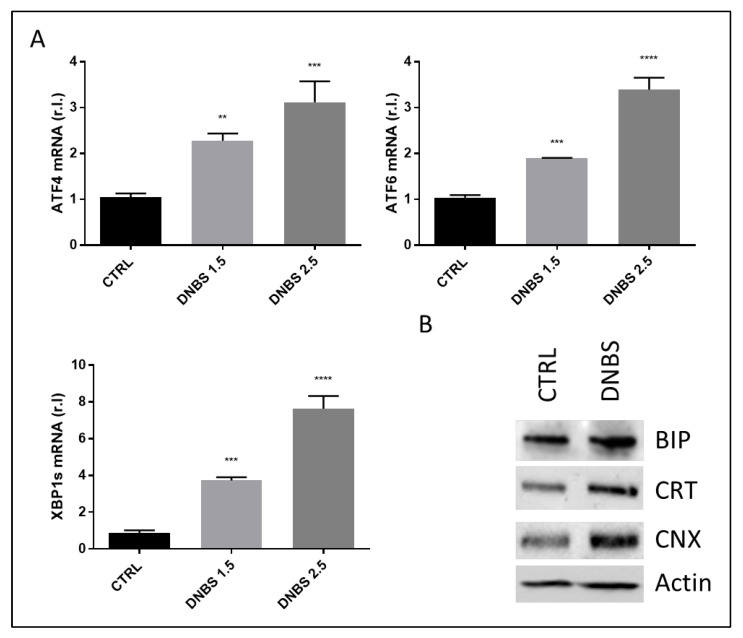
ER Stress induced by DNBS. Colon from C57BL/6 was cultivated for 5 h in a GEVS and untreated (CTRL) or treated with the indicated concentrations of DNBS and the expression levels of the ER stress markers ATF4, ATF6 or XBP1 were evaluated by qPCR (**A**). In parallel, the expression of BIP/Grp78, Calreticulin (CRT) and Calnexin (CNX) was evaluated by western blotting analysis in tissues exposed to 1.5 mg/mL DNBS or vehicle control (CTRL) (**B**). Each panel is representative of experiments performed in triplicate. Histograms represent mean ± SD of triplicate sample; **** *p* < 0.0001; *** *p* < 0.001; ** *p* < 0.01. Actin was used as loading control.

**Figure 4 biology-11-01574-f004:**
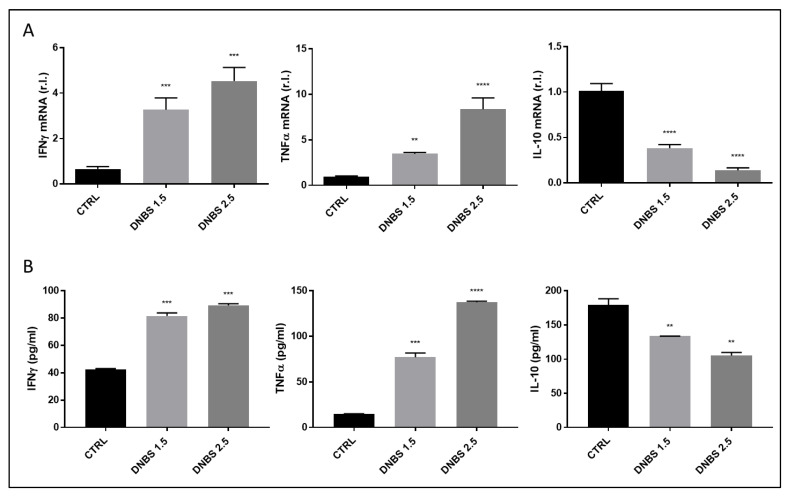
Inflammation induced by DNBS. Colon from C57BL/6 was cultivated for 5 h in a GEVS and untreated (CTRL) or treated with the indicated concentrations of DNBS and the expression levels of the pro-inflammatory cytokines IFNg and TNFa or the anti-inflammatory cytokine IL-10 were evaluated by both qPCR (**A**) and ELISA (**B**). Each panel is representative of experiments performed in triplicate. Histograms represent mean ± SD of triplicate sample; **** *p* < 0.0001; *** *p* < 0.001; ** *p* < 0.01.

**Figure 5 biology-11-01574-f005:**
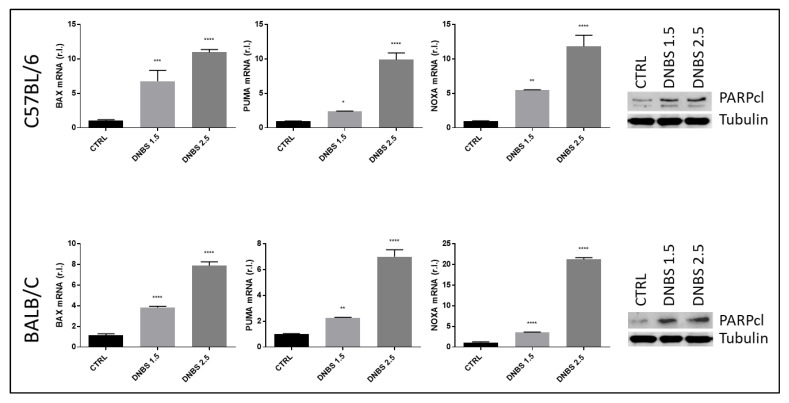
Apoptosis induction. Colon from C57BL/6 or BALB/c mice was cultivated for 5 h in a GEVS and untreated (CTRL) or treated with the indicated concentrations of DNBS and the expression levels of BAX and the two BH3-only proteins PUMA and NOXA, well-known pro-apoptotic markers, were evaluated by qPCR, while the cleavage of the caspases’ substrate PARP was evaluated by western blotting analysis on tissue lysates (right panels). Histograms represent mean ± SD of triplicate sample; **** *p* < 0.0001; *** *p* < 0.001; ** *p* < 0.01; * *p* < 0.1. Tubulin was used as loading control.

**Figure 6 biology-11-01574-f006:**
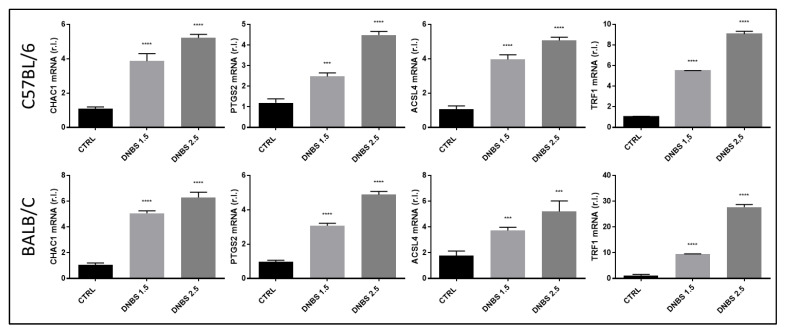
Ferroptosis induction. Colon from C57BL/6 (upper panels) or BALB/c (bottom panels) mice was cultivated for 5 h in a GEVS and untreated (CTRL) or treated with the indicated concentrations of DNBS and the expression levels of the well-known ferroptotic markers CHAC1, PTGS2, ACSL4 and TRF1 were evaluated by qPCR. Histograms represent mean ± SD of triplicate sample; **** *p* < 0.0001; *** *p* < 0.001.

**Figure 7 biology-11-01574-f007:**
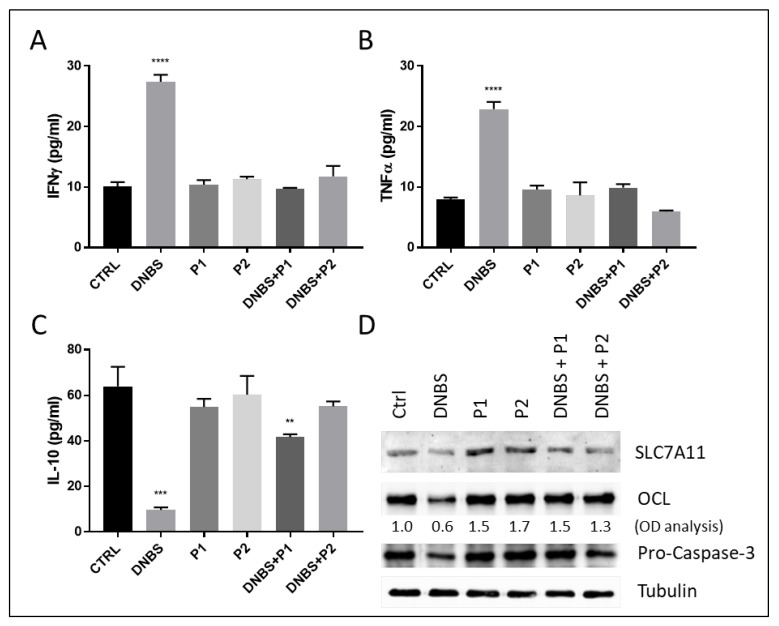
Impact of probiotics on DNBS-stimulated UC. Colon from C57BL/6 mice was untreated (CTRL) or exposed to DNBS (1.5 mg/mL), P1 or P2 alone or DNBS+P1 or DNBS + P2 for 5 h, in GEVS. The expression of the pro-inflammatory cytokines IFNg (**A**) and TNFa (**B**) or the anti-inflammatory IL-10 (**C**) was evaluated by ELISA on tissue lysates. (**D**) Protein levels of SLC7A11, OCL or pro-Caspase-3 was evaluated by western blotting analysis. Histograms represent mean ± SD of triplicate sample; **** *p* < 0.0001; *** *p* < 0.001; ** *p* < 0.01. Tubulin was used as loading control.

**Figure 8 biology-11-01574-f008:**
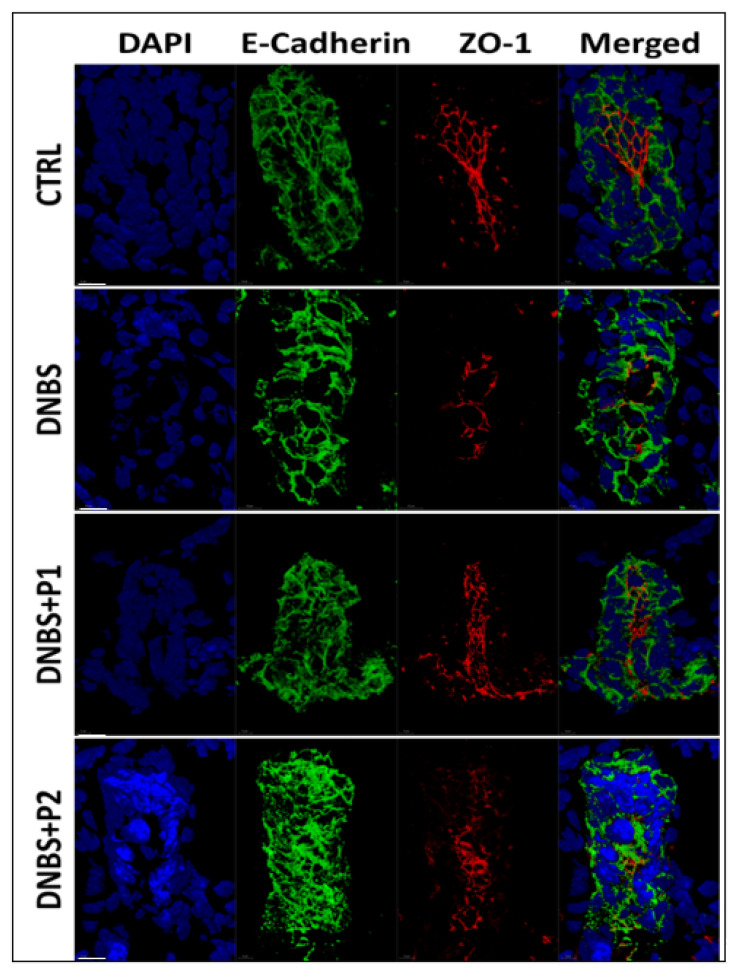
Impact of probiotics on DNBS-induced ZO-1 dysregulation. Colon from C57BL/6 mice was untreated (CTRL) or exposed to DNBS (1.5 mg/mL) alone or DNBS + P1 or DNBS + P2 for 5 h, in GEVS. Tissue distribution of TJs was evaluated analyzing the localization of ZO-1 (red). Epithelial cells were evidenced by the expression of E-Cadherin (green), while cell nuclei were evidenced by DAPI (blue) staining. Representative images of experiment performed in triplicate; scale bar = 10 mm.

**Table 1 biology-11-01574-t001:** Primer sequences.

ATF4_F	GTTTAGAGCTAGGCAGTGAAG
ATF4_R	CCTTTACACATGGAGGGATTAG
ATF6_F	GATGGTGACAACCAGAAAGA
ATF6_R	TGGAGGTGGAGGCATATAA
XBP1_F	AGTCCGCAGCAGGTG
XBP1_R	GGTCCAACTTGTCCAGAATG
CLD2_F	CCTCGCTGGCTTGTATTATC
CLD2_R	AAAGACTCCACCCACTACA
CLD15_F	GGGACCCTCCACATACTT
CLD15_R	CATACTTGGTTCCAGCATACA
OCL_F	TCTTTGGAGGAAGCCTAAAC
OCL_R	CTGCTCTTGGGTCTGTATATC
IFNγ_F	CCACATCTATGCCACTTGAG
IFNγ_R	CTCTTCCTCATGGCTGTTTC
TNFα_F	CCTATGTCTCAGCCTCTTCT
TNFα_R	GGGAACTTCTCATCCCTTTG
IL-10_F	TGAATTCCCTGGGTGAGA
IL-10_R	CCACTGCCTTGCTCTTATT
BAX_F	GGTTGCCCTCTTCTACTTTG
BAX_R	AGTGTCCAGCCCATGAT
PUMA_F	GGTTGCCCTCTTCTACTTTG
PUMA_R	AGTGTCCAGCCCATGAT
NOXA_F	GGTTGCCCTCTTCTACTTTG
NOXA_R	AGTGTCCAGCCCATGAT
CHAC1_F	TCACAGCACTGGCCTAT
CHAC1_R	CAAGGTTGTGACCAGAGAAG
PTGS2_F	GCCTGGTCTGATGATGTATG
PTGS2_R	GTCTGCTGGTTTGGAATAGT
ACSL4_F	TAAGCCCAXTTCAGACAAAC
ACSL4_R	GGCTACAGCATGGTCAAATA
TFR1_F	GGGCTATTGTAAGCGTGTAG
TFR1_R	CCTCTGTTTCCATGGTTTCT
GAPDH_F	TTCAACGGCACAGTCAAG
GAPDH_R	CCAGTAGACTCCACGACATA

## Data Availability

Data are contained within the article.

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
