# Peer review of "The Gut-Ex-Vivo System (GEVS) Is a Dynamic and Versatile Tool for the Study of DNBS-Induced IBD in BALB/C and C57BL/6 Mice, Highlighting the Protective Role of Probiotics"

_biology, 2022, doi:10.3390/biology11111574_

Round 1

Reviewer 1 Report (Previous Reviewer 2)

As I have already reviewed this manuscript and had minor comments the last time, the additions increase the novelty and interest in the GEVS system.

Author Response

We thank the reviewer for his/her positive comment on our work

Reviewer 2 Report (New Reviewer)

The manuscript is well documented, and each result is appropriately discussed. I have some comments which needs to be addressed before it is accepted:

1.     Immune infiltration plays an important role in IBD which is missing in this study.

2.    There are some reports which show that DSS or TNBS-mediated colitis down-regulates claudin-2 expression which contradicts this study. 

3.    Author is unable to show the relative expression of claudin-2 and claudin-15 in the proximal and basal colon.

4.    E-cadherin immunofluorescent staining show that colon histology is disrupted even in control. This shows that at baseline there is more necrosis and apoptosis.

5.    Inflammatory cytokines downregulate the E-cadherin expression but in this study, E-cadherin is upregulated.

Round 2

Reviewer 2 Report (New Reviewer)

Thanks for the clarification. it is an excellent manuscript. 

This manuscript is a resubmission of an earlier submission. The following is a list of the peer review reports and author responses from that submission.

Round 1

Reviewer 1 Report

In this article, the authors demonstrate that the set up of a gut ex-vivo system provides similar results to the in vivo DNBS-colitis. The work is very well written and evidence clearly stated and discussed. I wonder whether it is possible to add some other methods to confirm these results, other than PCR (eg immunofluorescence). By doing so, the interest for GEVS could even be higher. 

Reviewer 2 Report

The article titled "The Gut-Ex-Vivo System (GEVS) is a dynamic and versatile tool for the study of DNBS-induced IBD in BALB/C and C57BL/6 mice" utilizes a previously demonstrated system in a different murine model. The authors use mRNA and qPCR as their readout of cellular processes. The main issue with this is the novelty, the original GEVS was done on C57BL/6 mice, there fore this system has already been shown to work with tissue from these animals. While the GEVS system is unique and offers a great alternative to using multiple animals when doing a study, this particular manuscript does not offer any additional novelty to either the system or our greater knowledge of IBD.

Minor Comments

  1. Minor grammatical errors such as line 41 is missing some words
  2. Line 34 should be C7BL/6
  3. Figure 2 referenced in the text (Lines 186-188) should be figure 2A, B, C respectively

Major Comments

  1. The only readout the authors performed was qPCR, which is limited because it only shows one aspect of the cellular response. ELISA should be done on supernatants to see if there is also a change in cytokine secretion (IL-10, TNF, IFN etc)